# SCHEDULENET: LEARN TO SOLVE MINMAX MULTIPLE TRAVELLING SALESMAN PROBLEM

## ABSTRACT

There has been continuous effort to learn to solve famous CO problems such as Traveling Salesman Problem (TSP) and Vehicle Routing Problem (VRP) using reinforcement learning (RL). Although they have shown good optimality and computational efficiency, these approaches have been limited to scheduling a single-vehicle. *MinMax* mTSP, the focus of this study, is the problem seeking to minimize the total completion time for multiple workers to complete the geographically distributed tasks. Solving *MinMax* mTSP using RL raises significant challenges because one needs to train a distributed scheduling policy inducing the cooperative strategic routings using only the single delayed and sparse reward signal (makespan). In this study, we propose the ScheduleNet that can solve mTSP with any numbers of salesmen and cities. The ScheduleNet presents a state (partial solution to mTSP) as a set of graphs and employs type aware graph node embeddings for deriving the cooperative and transferable scheduling policy. Additionally, to effectively train the ScheduleNet with sparse and delayed reward (makespan), we propose an RL training scheme, Clipped REINFORCE with "target net," which significantly stabilizes the training and improves the generalization performance. We have empirically shown that the proposed method achieves the performance comparable to Google OR-Tools, a highly optimized meta-heuristic baseline.

## 1 INTRODUCTION

There have been numerous approaches to solve combinatorial optimization (CO) problems using machine learning. Bengio et al. (2020) have categorized these approaches into *demonstration* and *experience*. In *demonstration* setting, supervised learning has been employed to mimic the behavior of the existing expert (e.g., exact solvers or heuristics). On the other hand, in the *experience* setting, typically, reinforcement learning (RL) has been employed to learn a parameterized policy that can solve newly given target problems without direct supervision. While the demonstration policy cannot outperform its guiding expert, RL-based policy can outperform the expert because it improves its policy using a reward signal. Concurrently, Mazyavkina et al. (2020) have further categorized the RL approaches into *improvement* and *construction* heuristics. An *improvement* heuristics start from the arbitrary (complete) solution of the CO problem and iteratively improve it with the learned policy until the improvement stops (Chen & Tian, 2019; Ahn et al., 2019). On the other hand, the *construction* heuristics start from the empty solution and incrementally extend the partial solution using a learned sequential decision-making policy until it becomes complete.

There has been continuous effort to learn to solve famous CO problems such as Traveling Salesman Problem (TSP) and Vehicle Routing Problem (VRP) using RL-based construction heuristics (Bello et al., 2016; Kool et al., 2018; Khalil et al., 2017; Nazari et al., 2018). Although they have shown good optimality and computational efficiency performance, these approaches have been limited to only scheduling a single-vehicle. The *multi*-extensions of these routing problems, such as *multiple* TSP and *multiple* VRP, are underrepresented in the deep learning research community, even though they capture a broader set of the real-world problems and pose a more significant scientific challenge.

The multiple traveling salesmen problem (mTSP) aims to determine a set of subroutes for each salesman, given $m$ salesmen and $N$ cities that need to be visited by one of the salesmen, and a depot where salesmen are initially located and to which they return. The objective of a mTSP is either minimizing the sum of subroute lengths (*MinSum*) or minimizing the length of the longest subroute

(*MinMax*). In general, the MinMax objective is more practical, as one seeks to visit all cities as soon as possible (i.e., total completion time minimization). In contrast, the MinSum formulation, in general, leads to highly imbalanced solutions where one of the salesmen visits most of the cities, which results in longer total completion time (Lupoaie et al., 2019).

In this study, we propose a learning-based decentralized and sequential decision-making algorithm for solving Minmax mTSP problem; the trained policy, which is a construction heuristic, can be employed to solve mTSP instances with any numbers of salesman and cities. Learning a transferable mTSP solver in a construction heuristic framework is significantly challenging comparing to its single-agent variants (TSP and CVRP) because (1) we need to use the state representation that is flexible enough to represent any arbitrary number of salesman and cities (2) we need to introduce the coordination among multiple agents to complete the geographically distributed tasks as quickly as possible using a sequential and decentralized decision making strategy and (3) we need to learn such decentralized cooperative policy using only a delayed and sparse reward signal, makespan, that is revealed only at the end of the episode.

To tackle such a challenging task, we formulate mTSP as a semi-MDP and derive a decentralized decision making policy in a multi-agent reinforcement learning framework using only a sparse and delayed episodic reward signal. The major components of the proposed method and their importance are summarized as follows:

- Decentralized cooperative decision-making strategy: Decentralization of scheduling policy is essential to ensure the learned policy can be employed to schedule any size of mTSP problems in a scalable manner; decentralized policy maps local observation of each idle salesman one of feasible individual action while joint policy maps the global state to the joint scheduling actions.

- State representation using type-award graph attention (TGA): the proposed method represents a state (partial solution to mTSP) as a set of graphs, each of which captures specific relationships among works, cities, and a depot. The proposed method then employs TGA to compute the node embeddings for all nodes (salesman and cities), which are used to assign idle salesman to an unvisited city sequentially.

- Training decentralized policy using a single delayed shared reward signal: Training decentralized cooperative strategy using a single sparse and delayed reward is extremely difficult in that we need to distribute credits of a single scalar reward (makespan) over the time and agents. To resolve this, we propose a stable MARL training scheme which significantly stabilizes the training and improves the generalization performance.

We have empirically shown that the proposed method achieves the performance comparable to Google OR-Tools, a highly optimized meta-heuristic baseline. The proposed approach outperforms OR-Tools in many cases on in-training, out-of-training problem distributions, and real-world problem instances. We also verified that scheduleNet can provide an efficient routing service to customers.

## 2 RELATED WORK

**Construction RL approaches** A seminal body of work focused on the construction approach in the RL setting for solving CO problems (Bello et al., 2016; Nazari et al., 2018; Kool et al., 2018; Khalil et al., 2017). These approaches utilize *encoder-decoder* architecture, that encodes the problem structure into a hidden embedding first, and then autoregressively decodes the complete solution. Bello et al. (2016) utilized LSTM (Hochreiter & Schmidhuber, 1997) based encoder and decode the complete solution (tour) using Pointer Network (Vinyals et al., 2015) scheme. Since the routing tasks are often represented as graphs, Nazari et al. (2018) proposed an attention based encoder, while using LSTM decoder. Recently, Kool et al. (2018) proposed to use Transformer-like architecture (Vaswani et al., 2017) to solve several variants of TSP and single-vehicle CVRP. On the contrary, Khalil et al. (2017) do not use encoder-decoder architecture, but a single graph embedding model, structure2vec (Dai et al., 2016), that embeds a partial solution of the TSP and outputs the next city in the (sub)tour. (Kang et al., 2019) has extended structure2vec to random graph and employed this random graph embedding to solve identical parallel machine scheduling problems, the problem seeking to minimize the makespan by scheduling multiple machines.

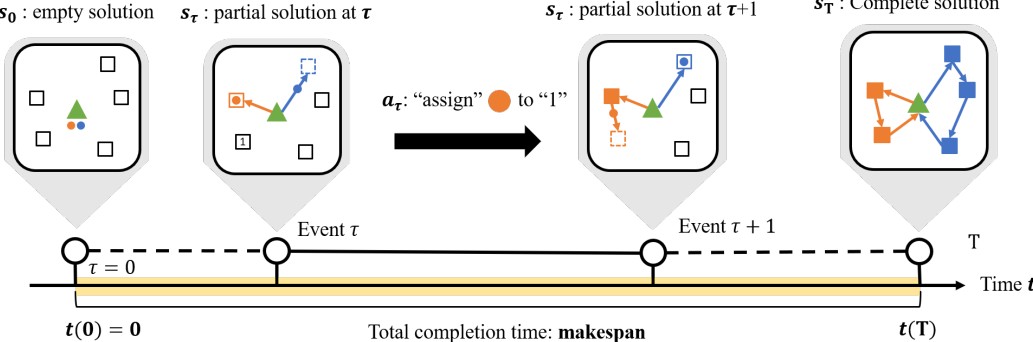

Figure 1: **The mTSP MDP** The black lined balls indicates the events of the mTSP MDP. The empty, dashed, and, filled rectangles represent the unassigned, assigned, and inactive cities, respectively. The circles represent the workers and the positions of the circles show the 2D coordinates of worker. The orange and blue colored lines shows the subtours of the orange and blue worker, respectively.

**Learned mTSP solvers** The machine learning approaches for solving mTSP date back to Hopfield & Tank (1985). However, these approaches require per problem instance training. (Hopfield & Tank, 1985; Wacholder et al., 1989; Somhom et al., 1999). Among the recent learning methods, Kaempfer & Wolf (2018) encodes *MinSum* mTSP with a set-specialized variant of Transformer architecture that uses permutation invariant pooling layers. To obtain the feasible solution, they use a combination of the softassign method Gold & Rangarajan (1996) and a beam search. Their model is trained in a supervised setting using mTSP solutions obtained by Integer Linear Programming (ILP) solver. Hu et al. (2020) utilizes a GNN *encoder* and self-attention Vaswani et al. (2017) policy outputs a probability of assignment to each salesman per city. Once cities are assigned to specific salesmen, they use existing TSP solver, OR-Tools (Perron & Furnon), to obtain each worker's sub-routes. Their method shows impressive scalability in terms of the number of cities, as they present results for mTSP instances with 1000 cities and ten workers. However, the trained model is not scalable in terms of the number of workers and can only solve mTSP problems with a pre-specified, fixed number of workers.

## 3 PROBLEM FORMULATION

We define the set of $m$ salesmen indexed by $\mathbb{V}_T = \{1, 2, ..., m\}$, and the set of $N$ cities indexed by $\mathbb{V}_C = \{m + 1, 2, ..., m + N\}$. Following mTSP conventions, we define the first city as the depot. We also define the 2D-coordinates of *entities* (salesmen, cities, and the depot) as $p^i$. The objective of MinMax mTSP is to minimize the length of the longest subtour of salesmen, while subtours covers all cities and all subtours of salesmen end at the depot. For the clarity of explanation, we will refer to salesman as a *workers*, and cities as a *tasks*.

### 3.1 MDP FORMULATION FOR MINMAX MTSP

In this paper, the objective is to construct an optimal solution with a construction RL approach. Thus, we cast the solution construction process of *MinMax* mTSP as a Markov decision process (MDP). The components of the proposed MDP are as follows.

**Transition** The proposed MDP transits based on *events*. We define an event as the the case where any worker reaches its assigned city. We enumerate the event with the index $\tau$ for avoiding confusion from the elapsed time of the mTSP problem. $t(\tau)$ is a function that returns the time of event $\tau$. In the proposed event-based transition setup, the state transitions coincide with the sequential expansion of the partial scheduling solution.

**State** Each entity $i$ has its own state $s_\tau^i = \left( p_\tau^i, \mathbb{1}_\tau^{\text{active}}, \mathbb{1}_\tau^{\text{assigned}} \right)$ at the $\tau$-th event. the coordinates $p_\tau^i$ is time-dependent for workers and static for tasks and the depot. Indicator $\mathbb{1}_\tau^{\text{active}}$ describes whether

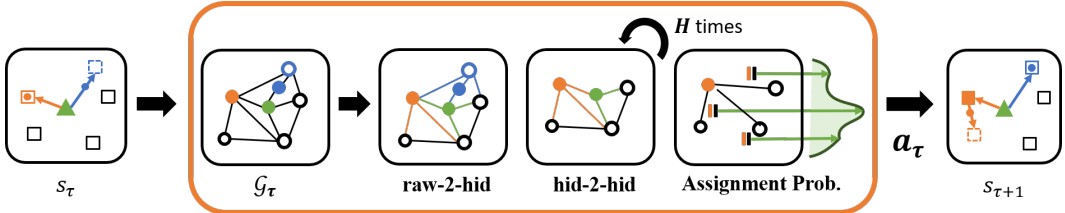

Figure 2: **Assignment action determination step of ScheduleNet**

the entity is *active* or *inactive* In case of tasks, inactive indicates that the task is already visited; in case of worker, inactive means that worker returned to the depot. Similarly, $\mathbb{1}_\tau^{\text{assigned}}$ indicates whether worker is assigned to a task or not. We also define the environment state $s_\tau^{\text{env}}$ that contains the current time of the environment, and the sequence of tasks visited by each worker, i.e., partial solution of the mTSP. The state $s_\tau$ of the MDP at the $\tau$-th event becomes $s_\tau = \left( \{ s_\tau^i \}_{i=1}^{m+N}, s_\tau^{\text{env}} \right)$. The first state $s_0$ corresponds to the empty solution of the given problem instance, i.e., no cities have been visited, and all salesmen are in the depot. The terminal state $s_\text{T}$ corresponds to a *complete* solution of the given mTSP instance, i.e., when every task has been visited, and every worker returned to the depot (See Figure 1).

**Action** A scheduling action $a_\tau$ is defined as the *worker-to-task assignment*, i.e. salesman has to visit the assigned city.

**Reward**. We formulate the problem in a delayed reward setting. Specifically, the sparse reward function is defined as $r(s_\tau) = 0$ for all non-terminal events, and $r(s_\text{T}) = t(\text{T})$, where T is the index of the terminal state. In other words, a single reward signal, which is obtained only for the terminal state, is equals to the makespan of the problem instance.

## 4 ScheduleNet

Given the MDP formulation for *MinMax* mTSP, we propose ScheduleNet that can recommend a scheduling action $a_\tau$ given the current state $\mathcal{G}_\tau$ represented as a graph, i.e., $\pi_\theta(a_\tau | \mathcal{G}_\tau)$. The ScheduleNet first presents a state (partial solution of mTSP) as a set of graphs, each of which captures specific relationships among workers, tasks, and a depot. Then ScheduleNet employs type-aware graph attention (TGA) to compute the node embeddings and use the computed node embeddings to determine the next assignment action (See figure 2).

### 4.1 Worker-Task Graph Representation

Whenever an event occurs and the global state $\mathbf{s}_\tau$ of the MDP is updated at $\tau$, ScheduleNet constructs a directed complete graph $\mathcal{G}_\tau = (\mathbb{V}, \mathbb{E})$ out of $\mathbf{s}_\tau$, where $\mathbb{V} = \mathbb{V}_T \cup \mathbb{V}_C$ is the set of nodes and $\mathbb{E}$ is the set of edges. We drop the time iterator $\tau$ to simplify the notations since the following operations only for the given time step. The nodes and edges and their associated features are defined as:

- $v_i$ denotes the node corresponding entity $i$ in mTSP problem. The node feature $x_i$ for $v_i$ is equal to the state $s_\tau^i$ of entity $i$. In addition, $k_i$ denote the type of node $v_i$. For instance, if the entity $i$ is *worker* and its $\mathbb{1}_\tau^{active} = 1$, then the $k_i$ becomes *active-worker* type.
- $e_{ij}$ denotes the edge between between source node $v_j$ and destination node $v_i$, representing the relationships between the two. The edge feature $w_{ij}$ is equal to the Euclidean distance between the two nodes.

### 4.2 Type-aware Graph Attention Embedding

In this section, we describe a *type-aware* graph attention (TGA) embedding procedure. We denote $h_i$ and $h_{ij}$ as the node and the edge embedding, respectively, at a given time step, and $h_i'$ and $h_{ij}'$ as the updated embedding by TGA embedding. A single iteration of TGA embedding consists of three phases: (1) edge update, (2) message aggregation, and (3) node update.

**Type-aware Edge update** Given the node embeddings $h_i$ for $v_i \in \mathbb{V}$ and the edge embeddings $h_{ij}$ for $e_{ij} \in \mathbb{E}$, ScheduleNet computes the updated edge embedding $h'_{ij}$ and the attention logit $z_{ij}$ as:

$$
\begin{aligned}
h'_{ij} &= \mathrm{TGA}_{\mathbb{E}}([h_i, h_j, h_{ij}], k_j) \\
z_{ij} &= \mathrm{TGA}_{\mathbb{A}}([h_i, h_j, h_{ij}], k_j)
\end{aligned}
\tag{1}
$$

where $\mathrm{TGA}_{\mathbb{E}}$ and $\mathrm{TGA}_{\mathbb{A}}$ are, respectively, the type-aware edge update function and the type-aware attention function, which are defined for the specific type $k_j$ of the source node $v_j$. The updated edge feature $h'_{ij}$ can be thought of as the message from the source node $v_j$ to the destination node $v_i$, and the attention logit $z_{ij}$ will be used to compute the importance of this message.

In computing the updated edge feature (message), $\mathrm{TGA}_{\mathbb{E}}$ and $\mathrm{TGA}_{\mathbb{A}}$ first compute the "type-aware" edge encoding $u_{ij}$, which can be seen as a dynamic edge feature varying depending on the source node type, to effectively model the complex type-aware relationships among the nodes. Using the computed "type-aware" edge encoding $u_{ij}$, these two functions then compute the updated edge feature and attention logit using a multiplicative interaction (MI) layer (Jayakumar et al., 2019). The use of MI layer significantly reduces the number of parameters to learn without discarding the expressibility of the embedding procedure. The detailed architecture for $\mathrm{TGA}_{\mathbb{E}}$ and $\mathrm{TGA}_{\mathbb{A}}$ are provided in Appendix A.4.

**Type-aware Message aggregation** The distribution of the node types in the mTSP graphs is highly imbalanced, i.e., the number of task-specific node types is much larger than the worker specific ones. This imbalance is problematic, specifically, during the message aggregation of GNN, since permutation invariant aggregation functions are akin to ignore messages from few-but-important nodes in the graph. To alleviate such an issue, we propose the following *type-aware* message aggregation scheme.

We first define the type $k$ neighborhood of node $v_i$ as the set of the $k$ typed source nodes that are connected to the destination node $v_i$, i.e., $\mathcal{N}_k(i) = \{v_l | k_l = k, \forall v_l \in \mathcal{N}(i)\}$, where $\mathcal{N}(i)$ is the in-neighborhood set of node $v_i$ containing the nodes that are connected to node $v_i$ with incoming-edges.

The node $v_i$ aggregates separately messages from the same type of source nodes. For example, the aggregated message $m_i^k$ from $k$-type source nodes is computed as:

$$
m_i^k = \sum_{j \in \mathcal{N}_k(i)} \alpha_{ij} h'_{ij}
\tag{2}
$$

where $\alpha_{ij}$ is the attention score computed using the attention logits computed before as:

$$
\alpha_{ij} = \frac{\exp(z_{ij})}{\sum_{j \in \mathcal{N}_k(i)} \exp(z_{ij})}
\tag{3}
$$

Finally, all aggregated messages per type are concatenated to produce the total aggregated message $m_i$ for node $v_i$ as

$$
m_i = \mathrm{concat}(\{m_i^k | k \in \mathbb{K}\})
\tag{4}
$$

**Type-aware Node update** The aggregated message $m_i$ for node $v_i$ is then used to compute the updated node embedding $h'_i$ using the type-aware graph node update function $\mathrm{TGA}_{\mathbb{V}}$ as:

$$
h'_i = \mathrm{TGA}_{\mathbb{V}}([h_i, m_i], k_i)
\tag{5}
$$

## 4.3 ASSIGNMENT PROBABILITY COMPUTATION

ScheduleNet model consists of two type-aware graph embedding layers that utilize the embedding procedure explained in the section above. The first embedding layer *raw-2-hid* is used to encode initial node and edge features $x_i$ and $w_{ij}$ of the (full) graph $\mathcal{G}_\tau$, to obtain initial hidden node and edge features $h_i^{(0)}$ and $h_{ij}^{(0)}$, respectively.

We define the *target subgraph* $\mathcal{G}_\tau^s$ as the subset of nodes and edges from the original (full) graph $\mathcal{G}_\tau$ that only includes a target-worker (unassigned-worker) node and all unassigned-city nodes. The

second embedding layer *hid-2-hid* embeds the target subgraph $\mathcal{G}_\tau^s$, $H$ times. In other words, a hidden node and edge embeddings $h_i^{(0)}$ and $h_{ij}^{(0)}$ are iteratively updated $H$ times to obtain final hidden embeddings $h_i^{(H)}$ and $h_{ij}^{(H)}$, respectively. The final hidden embeddings are then used to make decision regarding *the worker-to-task assignment*.

Specifically, probability of assigning target worker $i$ to task $j$ is computed as

$$y_{ij} = \text{MLP}_{actor}(h_i^{(H)}; h_j^{(H)}; h_{ij}^{(H)})$$
$$p_{ij} = \text{softmax}(\{y_{ij}\}_{j \in \mathbb{A}(\mathcal{G}_\tau)}) \tag{6}$$

where the $h_i^{(H)}$, and $h_{ij}^{(H)}$ is the final hidden node, edge embeddings, respectively. In addition, $\mathbb{A}(\mathcal{G}_\tau)$ denote the set of feasible actions defined as $\{v_j | k_j = \text{``Unassigned-task''} \forall j \in \mathbb{V}\}$.

## 5 TRAINING SCHEDULENET

In this section, we describe the training scheme of the ScheduleNet. Firstly, we explain reward normalization scheme which is used to reduce the variance of the reward. Secondly, we introduce a stable RL training scheme which significantly stabilizes the training process.

**Makespan normalization** As mentioned in Section 3.1, we use the makespan of mTSP as the only reward signal for training RL agent. We denote the makespan of given policy $\pi$ as $M(\pi)$. We observe that, the makespan $M(\pi)$ is a highly volatile depending on the problem size (number of cities and salesmen), the topology of the map, and the policy. To reduce the variance of the reward, we propose the following normalization scheme:

$$m(\pi, \pi_b) = \frac{M(\pi_b) - M(\pi)}{M(\pi_b)} \tag{7}$$

where $\pi$ and $\pi_b$ is the evaluation and baseline policy, respectively.

The normalized makespan $m(\pi, \pi_b)$ is similar to (Kool et al., 2018), but we additionally divide the performance difference by the makespan of the baseline policy, which further reduces the variance that is induced by the size of the mTSP instance.

From the normalized terminal reward $m(\pi, \pi_b)$, we compute the normalized return as follows:

$$G_\tau(\pi, \pi_b) := \gamma^{T-\tau} m(\pi, \pi_b) \tag{8}$$

where $T$ is the index of the terminal state, and $\gamma$ is the discount factor.

The normalized return $G_\tau(\pi, \pi_b)$ becomes smaller and converges to (near) zero as $\tau$ decreases. From the perspective of the RL agent, it allows to the agent to acknowledge *neutrality* of current policy compared to the baseline policy for the early phase of the MDP trajectory. It is natural since knowing the relative goodness of the policy is hard from the early phase of the MDP.

**Stable RL training** It is well known that the solution quality of CO problems, including the makespan of mTSP, is extremely sensitive to the action selection, and it thus prevents the stable policy learning. To address this problem, we propose the *clipped* REINFORCE, a variant PPO *without* the learned value function. We empirically found that it is hard to train the value function[1], thus we use normalized returns $G\tau(\pi_\theta, \pi_b)$ directly. Then, the objective of the clipped REINFORCE is given as follows:

$$\mathcal{L}(\theta) = \mathbb{E}_{\pi_\theta} \left[ \sum_{\tau=0}^{T} [min(\text{clip}(\rho_\tau, 1 - \epsilon, 1 + \epsilon)G\tau(\pi_\theta, \pi_b), \rho_\tau G\tau(\pi_\theta, \pi_b))] \right] \tag{9}$$

---

[1]Note that the value function is trained to predict the makespan of the state to serve as an advantage estimator. Due to the combinatorial nature of the mTSP, the target of value function, makespan, is highly volatile, which makes training value function hard. We further discuss this in the experiment section.

Table 1: **mTSP Uniform Results.** We report the mean and standard deviation of the makespans for different uniform maps. We evaluate 500 independent maps for each $N$ and $m$ pairs. The gaps between OR-Tools and ScheduleNet are measured per instances.

| | $N = 20$ | | | $N = 50$ | | |
|---|---|---|---|---|---|---|
| | $m = 2$ | $m = 3$ | $m = 5$ | $m = 5$ | $m = 7$ | $m = 10$ |
| OR-Tools | 2.500±0.24 | 2.063±0.22 | 1.820±0.27 | 2.076±0.22 | 1.974±0.27 | 1.951±0.29 |
| ScheduleNet | 2.751±0.27 | 2.255±0.24 | 1.948±0.28 | 2.372±0.23 | 2.176±0.26 | 2.098±0.28 |
| 2Phase-NI | 2.814±0.35 | 2.364±0.35 | 1.998±0.30 | 2.506±0.32 | 2.285 ± 0.31 | 2.111 ± 0.28 |
| 2Phase-FI | 3.028±0.37 | 2.540±0.37 | 2.090±0.33 | 2.712±0.33 | 2.429 ± 0.32 | 2.198 ± 0.30 |
| 2Phase-RI | 2.889±0.37 | 2.441±0.36 | 2.040±0.32 | 2.584±0.33 | 2.351 ± 0.31 | 2.144 ± 0.29 |
| 2Phase-NN | 2.963±0.41 | 2.456±0.38 | 2.028±0.32 | 2.611±0.33 | 2.348 ± 0.32 | 2.148 ± 0.29 |
| Gap (vs. OR-Tools) | 1.102±0.07 | 1.095±0.06 | 1.071±0.06 | 1.144±0.05 | 1.106±0.05 | 1.086±0.07 |

| | $N = 100$ | | | $N = 200$ | | |
|---|---|---|---|---|---|---|
| | $m = 5$ | $m = 10$ | $m = 15$ | $m = 10$ | $m = 15$ | $m = 20$ |
| OR-Tools | 2.373±0.18 | 2.001±0.30 | 2.015±0.31 | 2.145 ±0.24 | 2.302±0.27 | 2.383±0.61 |
| ScheduleNet | 2.881±0.19 | 2.230±0.26 | 2.161±0.31 | 2.496±0.53 | 2.383±0.61 | 2.444±0.57 |
| 2Phase-NI | 3.116±0.35 | 2.348±0.31 | 2.161±0.29 | 2.750 ± 0.30 | 2.447± 0.31 | 2.250± 0.29 |
| 2Phase-FI | 3.348±0.36 | 2.474±0.31 | 2.250±0.30 | 2.916 ± 0.32 | 2.569± 0.29 | 2.354± 0.29 |
| 2Phase-RI | 3.150±0.34 | 2.400±0.32 | 2.187±0.30 | 2.759 ± 0.31 | 2.477± 0.30 | 2.281± 0.30 |
| 2Phase-NN | 3.232±0.36 | 2.405±0.31 | 2.191±0.30 | 2.820 ± 0.30 | 2.504± 0.30 | 2.295± 0.29 |
| Gap (vs. OR-Tools) | 1.121±0.06 | 1.123±0.09 | 1.079±0.10 | 1.164 ±0.15 | 1.003±0.17 | 0.954±0.19 |

where

$$\rho_\tau = \frac{\pi_\theta(a_\tau | \mathcal{G}_\tau)}{\pi_{\theta_{old}}(a_\tau | \mathcal{G}_\tau)} \tag{10}$$

and $(\mathcal{G}_\tau, a_\tau) \sim \pi_\theta$ is the state-action marginal following $\pi_\theta$, and $\pi_{\theta_{old}}$ is the old policy.

**Training detail** We used the greedy version of current policy as the baseline policy $\pi_b$. After updating the policy $\pi_\theta$, we smooth the parameters of policy $\pi_\theta$ with the Polyak average (Polyak & Juditsky, 1992) to further stabilize policy training. The pseudo code of training and network architecture is given in Appedix A.5.1.

Table 2: **mTSPLib results.** The CPLEX results with ∗ are optimial solutions. Otherwise, the known-best upper bound of CPLEX results are reported (mTS). Meta-heuristic results are reproduced from Lupoaie et al. (2019). The results of the two leading heuristic algorithms are provided here. The full results are given in the appendix A.7.

| Instance | $m$ | CPLEX | OR-Tools | ScheduleNet | SOM | ACO | EA | 2Phase-NI | 2Phase-RI |
|---|---|---|---|---|---|---|---|---|---|
| *eil51* | 2 | 222.73 | 243.02 | 259.67 | 278.44 | 248.76 | 276.62 | 271.25 | 265.85 |
| | 3 | 159.57 | 170.05 | 172.16 | 210.25 | 180.59 | 208.16 | 202.85 | 195.89 |
| | 5 | 123.96 | 127.5 | 118.94 | 157.68 | 135.09 | 151.21 | 183.53 | 150.49 |
| | 7 | 112.07 | 112.07 | 112.42 | 136.84 | 119.96 | 123.88 | 129.65 | 127.72 |
| *berlin* | 2 | 4110.21 | 4665.47 | 4816.3 | 5350.83 | 4388.99 | 5038.33 | 5941.03 | 5785.00 |
| | 3 | 3244.37 | 3311.31 | 3372.14 | 4197.61 | 3468.9 | 3865.45 | 3811.49 | 4133.85 |
| | 5 | 2441.39 | 2482.57 | 2615.57 | 3461.93 | 2733.56 | 2853.63 | 2972.57 | 4108.58 |
| | 7 | 2440.92 | 2440.92 | 2576.04 | 3125.21 | 2510.09 | 2543.73 | 2972.57 | 2998.20 |
| *eil76* | 2 | 280.85 | 318 | 334.1 | 364.02 | 308.53 | 365.72 | 363.21 | 395.15 |
| | 3 | 197.34 | 212.41 | 226.54 | 278.63 | 224.56 | 285.43 | 302.1 | 276.58 |
| | 5 | 150.3 | 143.38 | 168.03 | 210.69 | 163.93 | 211.91 | 191.41 | 185.77 |
| | 7 | 139.62 | 128.31 | 151.31 | 183.09 | 146.88 | 177.83 | 173.81 | 155.54 |
| *rat99* | 2 | 728.75 | 762.19 | 789.98 | 927.36 | 767.15 | 896.72 | 916.55 | 890.86 |
| | 3 | 587.17 | 552.09 | 579.28 | 756.08 | 620.45 | 739.43 | 802.84 | 843.03 |
| | 5 | 469.25 | 473.66 | 502.49 | 624.38 | 525.54 | 596.87 | 668.6 | 675.39 |
| | 7 | 443.91 | 442.47 | 471.67 | 564.14 | 492.13 | 534.91 | 554.19 | 565.12 |
| Gap (vs. optimal) | | 1.00 | 1.03±0.06 | 1.08±0.06 | 1.31±0.06 | 1.09±0.03 | 1.24±0.10 | 1.30±0.11 | 1.31±0.14 |

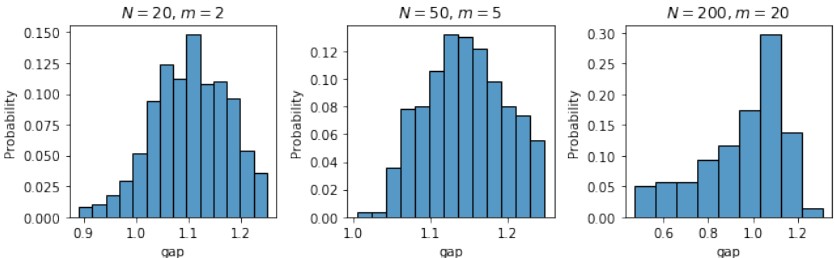

Figure 3: **Gap distributions of the random mTSP instances** [Left] Small-size random instance results, [Right] Medium-size random instance results, [Right] Large-size random instance results

## 6 EXPERIMENTS

We train the ScheduleNet using mTSP instances whose number $m$ of workers and the number $N$ of tasks are sampled from $m \sim U(2, 4)$ and $N \sim U(10, 20)$, respectively. This trained ScheduleNet policy is then evaluated on the various dataset, including randomly generated uniform mTSP datasets, mTSPLib (mTS), and randomly generated uniform TSP dataset, TSPLib, and TSP (dai). See Appendix for further training details.

### 6.1 MTSP RESULTS

**Random mTSP results** We firstly investigate the generalization performance of ScheduleNet on the randomly generated uniform maps with varying numbers of tasks and workers. We report the results of OR-Tools and 2Phase heuristics; 2Phase Nearest Insertion (NI), 2Phase Farthest Insertion (FI), 2Phase Random Insertion (RI), and 2Phase Nearest Neighbor (NN). The 2Phase heuristics construct sub-tours by (1) clustering cities with clustering algorithm, and (2) applying the TSP heuristics within the cluster. The details of implementation are provided in the appendix.

Table 1 shows that ScheduleNet in overall produces a slightly longer makespan than OR-Tools even for the large-sized mTSP instances. As the complexity of the target mTSP instance increases, the gap between ScheduleNet and OR-Tools decreases, even showing the cases where ScheduleNet outperforms OR-Tools. To further clarify, ScheduleNet has potentials for winning the OR-Tools on small and large cases as shown in the figure 3. This result empirically proves that ScheduleNet, even trained with small-sized mTSP instances, can solve large scale problems well. Notably, on the large scale maps, 2-Phase heuristics show their general effectiveness due to the uniformity of the city positions. It naturally invokes us to consider more realistic problems as discussed in the following section.

**mTSPLib results** The trained ScheduleNet is employed to solve the benchmark problems in mTSPLib, without additional training, to validate the generalization capability of ScheduleNet on unseen mTSP instances, where the problem structure can be completely different from the instances used during training. Table 2 compares the performance of the ScheduleNet to other baseline models, including CPLEX (optimal solution), OR-Tools, and other meta-heuristics (Lupoaie et al., 2019); self-organization Map (SOM), ant-colony Optimization (ACO), and evolutionary algorithm (EA). We report the best known upper-bound for CPLEX results whenever the optimal solution is not known.

OR-Tools generally shows promising results. Interestingly, OR-Tools also discovers the solution even better than the known upper-bounds. (e.g., eil76-$m$=5,7, rat99-$m$=5) That is possible for the large cases the search space of the exact method, CLPEX, becomes easily prohibitively large. Our method shows the second-best performance following OR-tools. The winning heuristic methods, 2Phase-NI/RI, shows drastic performance degradation on mTSPLib maps. It is noteworthy that our method, even in the zero-shot setting, performs better than the meta-heuristic methods, which perform optimization to solve each benchmark problem.

**Computational times** The mixed-integer linear programming (MILP) formulated mTSP problem becomes quickly intractable due to the exponential growth of search space, namely subtour elim-

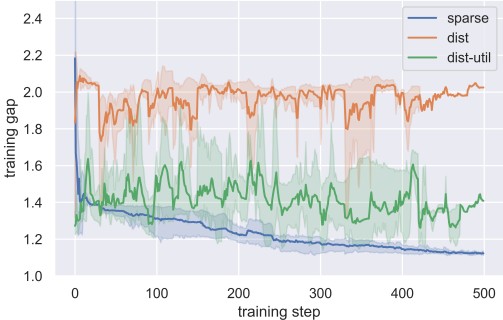 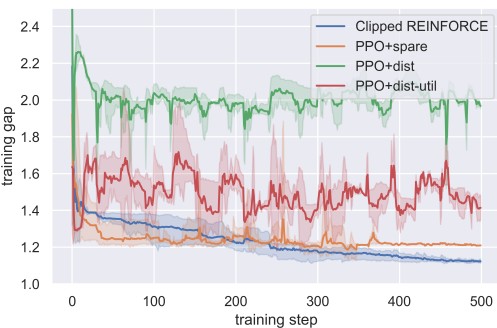

Figure 5: **[Left] Effect of reward function** The blue, orange, and green curve shows the training gap of the sparse reward, the *distance* reward, and *distance-utilization reward* over the training steps, respectively. The shadow regions visualize one standard deviation from the mean trends. We replicates 5 experiments per reward setup. **[Right] Effect of training method** The orange, green, and red curve shows the training gap of PPO model with the proposed spare, *distance*, and *distance-utilization* reward, respectively. We visualize the results of the clipped REINFORCE with the spare reward, which is denoted as blue curve, for clear comparisons. The shadow regions visualize one standard deviation from the mean trends. We replicates 5 experiments per setup.

ination constraint (SEC), as the number of workers increases. The computational gain of (Meta) heuristics, including the proposed method and OR-Tools, originates from the effective heuristics that trims out possible tours. The computational times of ScheduleNet linearly increase as the number of worker $m$ increases for the number for the fixed number of task $N$ due to the MDP formulation of mTSP. On the contrary, it is found that the computation times of OR-Tools depend on $m$ and $N$, and also graph topology. As a result, the ScheduleNet becomes faster than OR-Tools for large instances as shown in figure 6.

## 6.2 EFFECTIVENESS OF THE PROPOSED TRAINING SCHEME

Figure 5 compares the training curves of ScheduleNet and its variants. We firstly show the effectiveness of the proposed sparse reward compared to the dense reward functions; *distance* reward and *distance-utilization* reward. The *distance* reward is defined as the negative distance between the current worker position and the assigned city. This reward function is often used for solving TSP(Dai et al., 2016). The *distance-utilization* is defined as distance reward over the number of active workers. This reward function aims to minimize the (sub) tour distances while maximizing the utilization of the workers. The proposed sparse reward is the only reward function that can train ScheduleNet stable and achieves the minimal gaps, also as shown in 5 [Left].

We also validate the effectiveness of Clipped REINFORCE compared to the actor-critic counterpart, PPO. We use the same network architecture of the Clipped REINFORCE model for the actor and critic of the PPO model. Counter to the common belief, The actor-critic method (PPO) is not superior to the actor-only method (Clipped REINFORCE) as shown in 5 [Right]. We hypothesize this phenomenon is because the training target of the critic (sampled makepsan) is highly volatile and multi-modal as visualized in Figure 4 and the value prediction error would deteriorate the policy due to the bellman error propagation in actor-critic setup as discussed in Fujimoto et al. (2018).

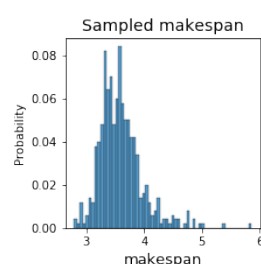

Figure 4: **makespan distribution**

## 7 CONCLUSION

We proposed ScheduleNet for solving *MinMax* mTSP, the problem seeking to minimize the total completion time for multiple workers to complete the geographically distributed tasks. The use of type-aware graphs and the specially designed TGA graph node embedding allows the trained ScheduleNet policy to induce the coordinated strate-

gic subroutes of the workers and to be well transferred to unseen mTSP with any numbers of workers and tasks. We have empirically shown that the proposed method achieves the performance comparable to Google OR-Tools, a highly optimized meta-heuristic baseline. All in all, this study has shown the potential that the proposed ScheduleNet can be effectively used to schedule multiple vehicles for solving large-scale, practical, real-world applications.

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

# A  APPENDIX

## A.1  DETAILS OF MDP TRANSITION AND GRAPH FORMULATION

**Event based MDP transition**  The formulated semi-MDP for ScheduleNet is event-based. Thus, whenever all workers are assigned to cities, the environment transits in time, until any of the workers arrives to the city (i.e. completes the task). Arrival of the worker to the city is the event trigger, meanwhile the other assigned workers are still on the way to their correspondingly assigned cities. We assume that each worker transits towards the assigned city with unit speed in the 2D Euclidean space, i.e. the distance travelled by each worker equals the time past between two consecutive MDP events.

**Graph formulation**  In total our graph formulation includes seven mutually exclusive node type: (1) assigned-worker, (2) unassigned-worker, (3) inactive-worker, (4) assigned-city, (5) unassigned-city, (6) inactive-city, and (7) depot. Here, the set of active workers (cities) is defined by the union of assigned and unassigned workers (cities). Inactive-city node refers to the city that has been already visited, while the inactive-worker node refers to the worker that has finished its route and returned to the depot.

## A.2  DETAILS OF IMPLEMENTATION

**2phase mTSP heuristics**  2phase heuristics for mTSP is an extension of well-known TSP heuristics to the $m > 1$ cases. First, we perform K-means spatial clustering of cities in the mTSP instance, where $K = m$. Next, we apply TSP insertion heuristics (Nearest Insertion, Farthest Insertion, Random Insertion, and Nearest Neighbour Insertion) for each cluster of cities. It should be noted that, performance of the 2phase heuristics is highly depended on the spatial distribution of the cities on the map. Thus 2phase heuristics perform particularly well on uniformly distributed random instances, where K-means clustering can obtain clusters with approximately same number of cities per cluster.

**Proximal Policy Optimization**  Our implementation of PPO closely the standard implementation of PPO2 from *stable-baselines* (Hill et al., 2018) with default hyperparameters, with modifications to allow for distributed training with Parameter Server.

## A.3  COMPUTATION TIME

Figure 6 shows the computation time curves as the function of number of cities (left), and number of workers (right). Overall, ScheduleNet is faster than OR-Tools, and the difference in computation speed only increases with the problem size. Additionally, ScheduleNet's computation time depends only on the problem size ($N + m$), whereas the computation time of OR-Tools on both the size of the problem and the topology of the underlying mTSP instance. In other words, the number of solutions searched by OR-Tools vary depending on the underlying problem.

Another computational and practical advantage of the ScheduleNet is its invariance to the number of workers. Computational complexity of ScheduleNet increases linearly with the number workers. On the other hand, the search space of meta-heuristic algorithms drastically increase with the number of workers, possibly, due to the exponentially increasing number Subtour Elimination Constraints (SEC). Particularly, we investigated that OR-Tools decreases the search space, by deactivating part of the workers, i.e. not utilizing all possible partial solutions (subtours). As a result, Figure 6 shows that computation time of the OR-Tools actually *decrease* due to deactivation part of workers, at the expense of the decreasing solution quality.

## A.4  DETAILS OF TYPE-AWARE GRAPH ATTENTION EMBEDDING

In this section, we thoroughly describe a *type-aware* graph embedding procedure. Similar to the main body, We overload notations for the simplicity of notation such that the input node and edge feature as $h_i$ and $h_{ij}$, and the embedded node and edge feature $h'_i$ and $h'_{ij}$, respectively.

The proposed graph embedding step consists of three phases: (1) type-aware edge update, (2) type-aware message aggregation, and (3) type-aware node update.

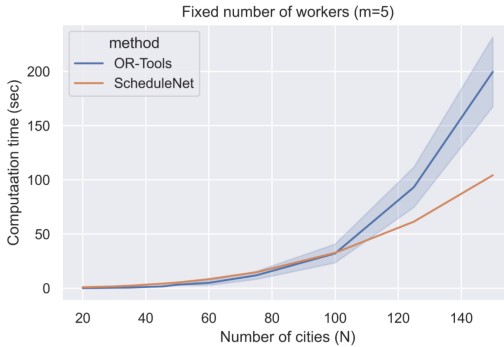 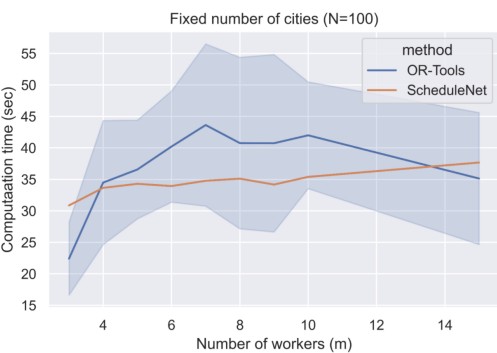

Figure 6: **Computation time of ScheduleNet and OR-Tools**. Figures show the computation time (in seconds) of ScheduleNet and OR-Tools as a function of number of cities (left) and number of workers (right). We sample 10 mTSP instances per $N$-$m$ combination for OR-Tools due to variability in the runtime due to underlying instance's topology. We measure all times on a CPU machine equip AMD Ryzen Threadripper 29990WX without any parallelization.

**Type-aware Edge update** The edge update scheme is designed to reflect the complex type relationship among the entities while updating edge features. First the *context* embedding $c_{ij}$ of edge $e_{ij}$ computed using the source node type $k_j$ such that:

$$c_{ij} = \text{MLP}_{etype}(k_j) \tag{11}$$

where $\text{MLP}_{etype}$ is the edge type encoder. The source node types are embedded into the context embedding $c_{ij}$ using $\text{MLP}_{etype}$. Next, the type-aware edge encoding $u_{ij}$ is computed using the Multiplicative Interaction (MI) layer (Jayakumar et al., 2019) as follows:

$$u_{ij} = \text{MI}_{edge}([h_i; h_j; h_{ij}], c_{ij}) \tag{12}$$

where $\text{MI}_{edge}$ is the edge MI layer. We utilize the MI layer, which dynamically generates its parameter depending on the context $c_{ij}$ and produces "type-aware" edge encoding $u_{ij}$, to effectively model the complex type relationships among the nodes. "type-aware" edge encoding $u_{ij}$ can be seen as a dynamic edge feature which varies depending on the source node type. After the updated edge embedding $h'_{ij}$ and its attention logit $z_{ij}$ is obtained as:

$$h'_{ij} = \text{MLP}_{edge}(u_{ij}) \tag{13}$$

$$z_{ij} = \text{MLP}_{attn}(u_{ij}) \tag{14}$$

where $\text{MLP}_{edge}$ and $\text{MLP}_{attn}$ is the edge updater and logit function, respectively. the edge updater and logit function produces updated edge embdding and logits from the "type-aware" edge.

The computation steps of equation 11, 12, and 13 are defined as $\text{TGA}_{\mathbb{E}}$. Similarly, the computation steps of equation 11, 12, and 14 are defined as $\text{TGA}_{\mathbb{A}}$.

**Message aggregation** First, we define the type-$k$ neighborhood of node $v_i$ such that $\mathcal{N}_k(i) = \{v_l | k_l = k, \forall v_l \in \mathcal{N}(i)\}$, where $\mathcal{N}(i)$ is the in-neigborhood set of node $i$. i.e., The type-$k$ neighborhood is the set of edges heading to node $i$, and their source nodes have type $k$. The proposed type-aware message aggregation procedure computes attention score $\alpha_{ij}$ for the $e_{ij}$, which starts from node $j$ and heads to node $i$, such that:

$$\alpha_{ij} = \frac{\exp(z_{ij})}{\sum_{l \in \mathcal{N}_{k_j}(i)} \exp(z_{il})} \tag{15}$$

Intuitively speaking, The proposed attention scheme normalizes the attention logits of incoming edges over the types. Therefore, the attention scores sum up to 1 over each type-$k$ neighborhood.

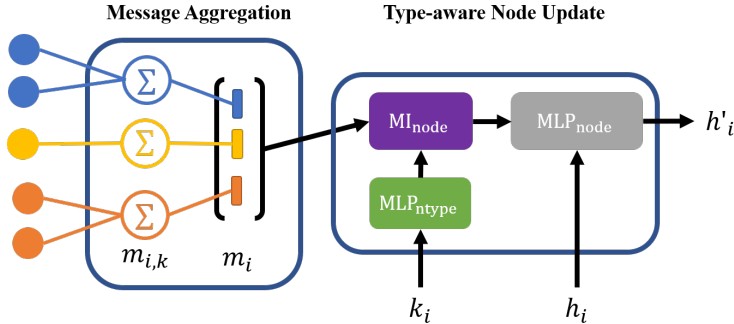

Figure 7: **Type-aware graph attention embedding** We omit the type-aware edge update for the clarity of visualization.

Next, the type-$k$ neighborhood message $m_{i,k}$ for node $v_i$ is computed as:

$$m_i^k = \sum_{j \in \mathcal{N}_k(i)} \alpha_{ij} h'_{ij} \tag{16}$$

In this aggregation step, the incoming messages of node $i$ are aggregated type-wisely. Finally, all incoming type neighborhood messages are concatenated to produce (inter-type) aggregated message $m_i$ for node $v_i$, such that:

$$m_i = \text{concat}(\{m_i^k | k \in \mathbb{K}\}) \tag{17}$$

**Node update** Similar to the edge update phase, first, the context embedding $c_i$ is computed for each node $v_i$:

$$c_i = \text{MLP}_{ntype}(k_i) \tag{18}$$

where $\text{MLP}_{ntype}$ is the node type encoder. Then, the updated hidden node embedding $h'_i$ is computed as below:

$$h'_i = \text{MLP}_{node}(h_i, u_i) \tag{19}$$

where $u_i = \text{MI}_{node}(m_i, c_i)$ is the type-aware node embedding that is produced by $\text{MI}_{node}$ layer using aggregated messages $m_i$ and the context embedding $c_i$.

The computation steps of equation 18, and 19 are defined as $\text{TGA}_{\mathbb{E}}$. The overall computation procedure TGA is illustrated in Figure 7.

### A.5 DETAILS OF SCHEDULENET TRAINING

#### A.5.1 TRAINING PSEUDO CODE

In this section, we presents a pseudocode for training ScheduleNet.

---

**Algorithm 1:** ScheduleNet Training

---

**Input:** Training policy $\pi_\theta$
**Output:** Smoothed policy $\pi_\phi$

1. Initialize smoothed policy with parameters $\phi \leftarrow \theta$.
2. **for** *update step* **do**
3.     Generate a random mTSP instance $I$
4.     **for** *number of episodes* **do**
5.         Construct mTSP MDP from the instance $I$
6.         $\pi_b \leftarrow \arg\max(\pi_\theta)$
7.         Collect samples with $\pi_\theta$ and $\pi_b$ from the mTSP MDP.
8.     $\pi_{\theta\,\text{old}} \leftarrow \pi_\theta$
9.     **for** *inner updates K* **do**
10.         $\theta \leftarrow \theta + \alpha \nabla_\theta \mathcal{L}(\theta)$
11.     $\phi \leftarrow \beta\phi + (1 - \beta)\theta$

---

A.5.2 HYPERPARAMETERS

In this section, we fully explain hyperparameters of ScheduleNet.

**Network Architecture** We use the same hyperparmeters for the *raw-2-hid* TGA layer and the *hid-2-hid* TGA layer. $MLP_{\text{etype}}$ and $MLP_{\text{ntype}}$ has one hidden layer with 32 neurons and their output dimensions are both 32. Both MI layers has 64 dimensional outputs. $MLP_{\text{edge}}$, $MLP_{\text{attn}}$, and $MLP_{\text{node}}$ has 2 hidden layers with 32 neurons. $MLP_{\text{actor}}$ has 2 hidden layers and the hidden layers has 128 neurons each. We use ReLU activation functions for all hidden layers. The hidden graph embedding step $H$ is two.

**Training** We use the discount factor $\gamma$ of 0.7. We use Adam (Kingma & Ba, 2014) with learning rate value of 0.001. We set the clipping parameter $\epsilon$ as 0.2. We sample 40 independent mTSP trajectories per gradient update. We clipped the maximum gradient whenever the norm of gradient is larger than 0.5. Inner updates steps K is three. The smoothing parameter $\beta$ is 0.95.

A.6 TRANSFERABILITY TEST ON TSP ($m = 1$)

The trained ScheduleNet has been employed to solve random TSP instances. Because ScheduleNet can be used to schedule any $m$ number of workers, if we set $m = 1$, it can be used to schedule TSP instance without further training. Table 3 shows the results on this transferability experiments.

Table 3 shows that the trained ScheduleNet can solve reasonably well on random TSP instances, although ScheduleNet has never been exposed to such TSP instances. Note that as the size of TSP increases, the gap between the ScheduleNet and other models becomes smaller. If the ScheduleNet is trained with TSP instances with $m = 1$, the performance can be further improved. However, we did not try that experiment to check its transferability over different types of routing problems with different objectives.

Table 3: Performance comparison on random uniform TSP instances. The Obj. defines the makespan, i.e., the length of the tour of the salesman.

| Method | $N = 20$ | | | $N = 50$ | | | $N = 100$ | | |
|---|---|---|---|---|---|---|---|---|---|
| | Obj. | Gap | Time | Obj. | Gap | Time | Obj. | Gap | Time |
| Concorde | 3.84 | 0.00% | (1m) | 5.70 | 0.00% | (2m) | 7.76 | 0.00% | (3m) |
| LKH3 | 3.84 | 0.00% | (18s) | 5.70 | 0.00% | (5m) | 7.76 | 0.00% | (21m) |
| Gurobi | 3.84 | 0.00% | (7s) | 5.70 | 0.00% | (2m) | 7.76 | 0.00% | (17m) |
| Nearest Insertion | 4.33 | 12.91% | (1s) | 6.78 | 19.03% | (2s) | 9.46 | 21.82% | (6s) |
| Farthest Insertion | 3.93 | 2.36% | (1s) | 6.01 | 5.53% | (2s) | 8.35 | 7.59% | (7s) |
| OR-Tools | 3.85 | 0.37% | (1s) | 5.80 | 1.83% | (2s) | 7.99 | 2.90% | (7s) |
| Bello et al. (2016) | 3.89 | 1.42% | – | 5.95 | 4.46% | – | 8.30 | 6.90% | – |
| Khalil et al. (2017) | 3.89 | 1.42% | – | 5.99 | 5.16% | – | 8.31 | 7.03% | – |
| AM (greedy) | 3.85 | 0.34% | (0s) | 5.80 | 1.76% | (2s) | 8.12 | 4.53% | (6s) |
| Am (sampling) | 3.84 | 0.08% | (5m) | 5.73 | 0.52% | (24m) | 7.94 | 2.26% | (1h) |
| ScheduleNet | 4.01 | 4.34% | (2s) | 6.11 | 7.01% | (9s) | 8.71 | 12.35% | (34s) |

A.7 EXTENDED MTSPLIB RESULTS

| | $m$ | CPLEX | OR | SN | Meta-heuristics | | | | Heuristics | | | |
| --- | --- | --- | --- | --- | SOM | ACO | EA | 2phase NI | 2phase FI | 2phase RI | 2phase NN |
| eil51 | 2 | 222.73 | 243.02 | 259.67 | 278.44 | 248.76 | 276.62 | 271.25 | 311.26 | 265.85 | 387.20 |
| | 3 | 159.57 | 170.05 | 172.16 | 210.25 | 180.59 | 208.16 | 202.85 | 218.71 | 195.89 | 222.13 |
| | 5 | 123.96 | 127.5 | 118.94 | 157.68 | 135.09 | 151.21 | 183.53 | 180.21 | 150.49 | 210.75 |
| | 7 | 112.07 | 112.07 | 112.42 | 136.84 | 119.96 | 123.88 | 129.65 | 144.11 | 127.72 | 147.81 |
| berlin | 2 | 4110.21 | 4665.47 | 4816.3 | 5350.83 | 4388.99 | 5038.33 | 5941.03 | 6605.05 | 5785.00 | 6519.51 |
| | 3 | 3244.37 | 3311.31 | 3372.14 | 4197.61 | 3468.9 | 3865.45 | 3811.49 | 4037.10 | 4133.85 | 3581.21 |
| | 5 | 2441.39 | 2482.57 | 2615.57 | 3461.93 | 2733.56 | 2853.63 | 2972.57 | 4037.10 | 4108.58 | 3581.21 |
| | 7 | 2440.92 | 2440.92 | 2576.04 | 3125.21 | 2510.09 | 2543.73 | 2972.57 | 3033.00 | 2998.20 | 3198.18 |
| eil76 | 2 | 280.85 | 318 | 334.1 | 364.02 | 308.53 | 365.72 | 363.21 | 403.56 | 395.15 | 373.75 |
| | 3 | 197.34 | 212.41 | 226.54 | 278.63 | 224.56 | 285.43 | 302.1 | 279.33 | 276.58 | 357.77 |
| | 5 | 150.3 | 143.38 | 168.03 | 210.69 | 163.93 | 211.91 | 191.41 | 204.16 | 185.77 | 197.54 |
| | 7 | 139.62 | 128.31 | 151.31 | 183.09 | 146.88 | 177.83 | 173.81 | 172.94 | 155.54 | 161.36 |
| rat99 | 2 | 728.75 | 762.19 | 789.98 | 927.36 | 767.15 | 896.72 | 916.55 | 965.94 | 890.86 | 929.82 |
| | 3 | 587.17 | 552.09 | 579.28 | 756.08 | 620.45 | 739.43 | 802.84 | 802.88 | 843.03 | 809.90 |
| | 5 | 469.25 | 473.66 | 502.49 | 624.38 | 525.54 | 596.87 | 668.6 | 645.91 | 675.39 | 641.89 |
| | 7 | 443.91 | 442.47 | 471.67 | 564.14 | 492.13 | 534.91 | 554.19 | 577.00 | 565.12 | 504.71 |
| gap | | 1 | 1.03 | 1.08 | 1.31 | 1.09 | 1.24 | 1.30 | 1.38 | 1.31 | 1.40 |

