# OpenReview forum: "ScheduleNet: Learn to Solve MinMax mTSP Using Reinforcement Learning with Delayed Reward"
_ICLR.cc/2021/Conference — Reject_

### Official Review · AnonReviewer3 · 2020-10-28
**An interesting RL framework for the mTSP problem with limited mild empirical results**

**Rating:** 5
**Confidence:** 4

**Review:**

Summary
-------------
The paper proposes a reinforcement learning approach to solve the min-max multiple TSP, where there are multiple salesmen and the goal is to minimize the longest subtour while every city is visited by one salesman. The authors propose an architecture, ScheduleNet, that encodes a partial solution or state and outputs a policy, i.e. a probability distribution over the actions. They train the model using a variant of the REINFORCE algorithm. The approach is validated on randomly generated mTSP instances as well as the standard literature benchmark TSPlib.


Strong points
-------------------
1. The addressed problem, TSP with multiple salesmen is an important combinatorial problem that is more challenging than the standard TSP because of the multi-agent cooperation that it involves. It is true that although there is a lot of literature on learning-based approaches that solve the TSP, only a few very recent papers deal with the multiple agent setting
2. The MDP formulation with the notion of events is sound and clearly explained. It is more sophisticated than the standard MDPs used in the “one agent” setting
3. The type-aware embeddings are interesting here to differentiate the interactions between the different types of nodes.


Weak points
-----------------
4. The numerical experiments are not convincing me that the approach would be useful in practice
5. To be informative, results in Table 1 should be the average over a number of random instances for each characteristic. Maybe it’s already the case but it is not mentioned. Moreover, the random instances should also follow different distributions to be varied and really helpful to evaluate the method.
6. Table 1 and 2: I found the reported gap (fraction of objectives) not so clear to get a precise sense of the performance. In Table 2, using the standard (approximate) optimiality gap would be better (obj_heuristic – obj_cplex)/obj_cplex.
7. It would be informative to report CPLEX results for the randomly generated instances as well.
8. Although the TSP is a natural special case of mTSP, the performance on of the approach on randomly generated TSP instances (cf Table 3) is significantly poorer than that of other learned heuristics. It would be useful to report the results for TSPlib as well.
9. The authors claim that they propose a new approach for training “Clipped REINFORCE, a variant of clipped PPO without the learned value function”. It would be useful to give more explanations for this choice.
10. In equation (9), I believe there is a missing sum over \tau. This does not help in understanding


Recommendation
-------------------------
I would vote for reject. In summary, the proposed approach is an adaptation of known techniques, to a specific interesting problem, that does not lead to a clear gain in performance.

Arguments for recommendation
---------------------------------------------
11. The MDP framework and type-aware GNNs are interesting and new in this context but not novel
12. To me, the numerical experiments are very limited and do not demonstrate the added-value of this method, see weak points above


Questions to authors
-----------------------------
13. Sec 4.1: It is said that you consider the complete graph and that the edge features are the Euclidian distance which is symmetric. So what is the point of using a *directed* graph?
14. Sec 4.1: “v_i denotes the node corresponding entity i in mTSP problem”. It sounds like v_i is a node of the graph. But if at \tau a worker is in between two cities, what would be v_i?
15. Sec 4.1: what are the types exactly? You give an example “active-worker” but it would be useful to list them all.
16. Sec 4: There is a confusion between source and destination indices. “eij denotes the edge between between source node vi and destination node vj” but then for the edge embedding “the specific type kj of the source node vj” and “the message from the source node vj to the destination node vi”. Similarly, equation (2), it is confusing to use j as a source index.
17. Sec 4.2: the definition of Nk(i) = {vj |kj = k, ∀l ∈ N (i)} does not make sense. What is the correct one? Because the graph is complete, is N(i) different from the entire V?
18. Sec 5: “\pi_b is the evaluation and baseline policy”. What baseline did you use?
19. Sec 5: equation 8, can you explain the choice of the exponent of gamma?
20. Sec 6:  “m ∼ U(2, 4) and N ∼ U(10, 20)”. Are m and N switched here? Otherwise there would be more workers than cities.
21. Table 2: what are SOM, ACO and EA? These baselines should be described (at least named) in the text.

Feedback to help improve the paper
--------------------------------------------------
22. “For the clarity of explanation, we will refer to salesman as a workers, and cities as a tasks.” I actually found it more confusing. Especially because task is standardly used to refer to the entire problem that the RL algorithm is addressing
23. “We define the set of m salesmen VT = {1, 2, ..., m}, and the set of N cities VC = {m+1, 2, ..., m+ N}” -> set of m salesmen *indexed by* VT = {1, 2, ..., m}, and the set of N cities *indexed by* VC = {m+1, 2, ..., m+ N}
24. “CPLEX results are reported as the average of the upper and lower bound”. It would make more sense to report the upper bound, i.e. the value of the best feasible solution found by the solver within the time limit.
25. To be able to better generalize to the TSP instances, why not include instances with N=1 during training.

---

> ### Author Response · Authors · 2020-11-24
> **Response to the reviewer 3**
>
> Thanks for the review. We revised the manuscript such that the questions are well answered. we have modified the introduction to further highlight the motivation and novelties of the current study. Please check the updated manuscript.
>
> Answers for questions 13: The proposed TGA utilize the directional information severely. We empirically confirmed that (even though it is not discussed in the manuscript) aggregating the messages depending on the source helps the agent each better performance. It is naturally make sense that the edge from "task" to "worker" has different meaning for the gnn and as well as the RL agent  to the edge from "worker" to "task".
>
> Answers for question 14: We consider the unit travel time for all workers.  i.e., If the worker nods $v_i$ is on the way to the assigned city, the worker's position $i$ will be on some point of the line connecting its source city, and the destination (assigned) city. We additionally explains the state update mechanism (which is handled by the simulator) in the appendix.
>
> Answers for question 15: "Inactive-worker, Assigned-worker, Unassigned-worker
> Inactive-task, Assigned-task, Unassigned-task, and Depot" are the list of the node types. We also update the appendix to give the full list of the types of nodes.
>
> Answer for question 16: Thank you for commenting on critical point for enhancing readability.  For the revised manuscript, Any edge related quantities such as $e_{ij}$, $h_{ij}$, $\alpha_{ij}$, $z_{ij}$ use the unified notation rules such that $j$ is the source node and the $i$ is the destination node.
>
> Answers for question 17: Yes, we realized that the notation can be misreading. The k-type neighborhood set $\mathcal{N}_k(i)$ is for enumerating the nodes which are in-neighborhood of node $i$ and simultaneously type $k$ (i.e., $k_j=k$).
>
> Answer for question 18: We use the greedy version of the current policy as the baseline policy $\pi_b$. The reason we choose to use the greedy version is two-fold (1) It can be seen as self-improving targets as discussed in [1],[2] (2) We consider practical cases where the baseline policy is not applicable easily. For instance, Job-shop-scheduling problems (JSSP) is known to hard for solving the problems but practically demanding in modern manufacturing facilities such as semi-conductor fabrication systems. Since the problem itself is hard, it is hard to get "nice" baseline, especially when considering computational budgets.
>
> Answer for question 19: The discount factor $\gamma$ is used to introducing the "neutrality" concept while training policy. The normalized makespan and the normalized return become (near) zero whenever the current training policy is neither better nor worse than the baseline policy. The normalized return will be (nearly) zero during the early phase of the mTSP MDP, which is intended since it is hard to know which policy is better during the early phase.
>
> Answer for question 20: Yes, it should be vice-versa. We revise the manuscript.
>
> Answer for question 21: Self-organization Map (SOM), Ant Colony Optimization (ACO), and Evolutionary Algorithms (EA) are one of the most well studied and practically favored meta-heuristic algorithms. We choose to such meta-heuristic algorithms as baseline since they often leads nice performances within moderate computational burdens.  We majorly update the manuscripts to give more explanation and comparisons toward the meta-heuristic algorithms.
>
> Answer for question 22: We agree that, in the field of RL, tasks reflects the objective of RL agents especially in meta-RL cases. However, we hope to keep the terminology "task" because, in various scheduling problems, the tasks are often used to describe any quantities to process; for instance, in JSSP, the "task" is used to describe the single step of manufacturing process.
>
> Answer for question 24: We report the upper bounds instead of the average of lower and upper bounds.
>
> [1] Mastering the game of Go with deep neural networks and tree search
> [2] Attention, Learn to Solve Routing Problems!

---

> > ### Author Response · Authors · 2020-11-25
> > **Comments for the weak points.**
> >
> > Throughout the rebuttal, we further analyze the proposed approaches, and we wish the additional materials can show the potential of RL methods for solving the multi-worker scheduling problem.
> >
> > [The numerical experiments are not convincing me that the approach would be useful in practice]
> > We agree that, at first, our numerical results might be seen as weak. We add more experiments and supporting arguments on the updated manuscript for showing the practicality in the perspective of achieving better makespan and shortening the computational times.
> > We want to highlight a few updates as follows:
> > Our method is not losing-algorithm. From the gap (makespan of ScheduleNet/ makespan of OR-Tools), distributions, we show our method can discover a better solution than OR-Tools in any size random uniform maps.
> > Our method is faster than OR-Tools. Due to the proposed MDP formulation of mTSP, our method can solve the given mTSP instance than OR-tools. The time gap between the two methods becomes severe as the number of cities increase. Please kindly see Section 6.1 and Figure 6 in the appendix.
> > Our method becomes "practical" when the scheduling decisions are required to be frequently made from two new updated findings, such as job dispatching in automated warehouse facilities.
> >
> > [To be informative, results in Table 1 should be the average over a number of random instances for each characteristic. Maybe it’s already the case but it is not mentioned. Moreover, the random instances should also follow different distributions to be varied and really helpful to evaluate the method.]
> >
> > First of all, we apologize for the confusion. As you pointed out, the metrics which were on the previous table 1 were computed as you anticipated.
> > However, we redesign the table 1 to become more statistically valid. We report the makespans of the given method by computing the mean and standard deviation for each pair of $N$ and $m$. The gaps, which show the relative goodness of the proposed model compared to the OR-Tools, are measured per instance.
> >
> > [Table 1 and 2: I found the reported gap (fraction of objectives) not so clear to get a precise sense of the performance. In Table 2, using the standard (approximate) optimiality gap would be better (obj_heuristic – obj_cplex)/obj_cplex.]
> >
> > We agree to report the gap compared to the "optimal solution" is better than the "OR-Tools" gap. However, due to the computational intractability of mTSP problems, it was unavailable to solve the large cases (e.g., $N$=200 $m$=5) within the appropriate time. The premature termination of the optimal solver (CPLEX) would result in too poor outcomes, and all the performance metrics could be optimistically biased.
> >
> > Hence we decided to use the OR-Tools, known for reliably working in general cases with less performance variability across the mTSP instance, as the target. We also found that OR-Tools achieve near-optimal results for some benchmark problems.
> >
> > [It would be informative to report CPLEX results for the randomly generated instances as well.]
> >
> > [The authors claim that they propose a new approach for training “Clipped REINFORCE, a variant of clipped PPO without the learned value function”. It would be useful to give more explanations for this choice.]
> >
> > The combinatorial optimization problems, including mTSP, is notorious for behaving in a highly volatile manner. For instance, the very first action selection of mTSP MDP would change the consequent trajectory in total. And most of the cases similar state cannot be seen again in the given mTSP trajectory. This translates into makespan variability.
> >
> > Following our return definition, if one uses an actor-critic setup, the target of the critic is some (discounted) statistics transformed from the makespan. It naturally leads the hardship in training value functions and results in inaccurate value prediction of the critic.
> >
> > We also confirmed that training value function. (i.e., PPO) is not helpful for achieving a better scheduler. Clipped REINFORCE shows better performance than PPO. We update the manuscript to better explains this phenomenon with additional experiments. Please kindly refer to the updated section 6.2 and Figure 4,5.
> >
> > [In equation (9), I believe there is a missing sum over \tau. This does not help in understanding]
> > Thanks for the comments. We update the main body such that the summation over the event index is considered in the objective function.

---

### Official Review · AnonReviewer2 · 2020-10-28
**Good initial results, but not enough at this point.**

**Rating:** 4
**Confidence:** 5

**Review:**

Summary of the paper:
This paper proposes a deep reinforcement learning (DRL) approach for learning a solution strategy for the minimum-makespan multiple Traveling Salesman Problem (mTSP). The makespan mTSP is a challenging combinatorial optimization problem in which we are given the 2-dimensional locations of a set of customers that must be visited by a (much smaller) set of trucks. The trucks depart from the same depot, and must return to it after their tours. The minimum makespan version of mTSP asks for a set of such tours such that the length of the longest tour is minimized.

This work is part of a recent interest in using machine learning to design algorithms for hard discrete optimization problems. A number of such methods have been proposed for the standard TSP problem, but the minimum-makespan mTSP brings a number of challenges: the makespan is a sparse reward signal in RL terms, in that it is realized only after a full solution has been constructed (at the end of the episode); unlike the TSP, the mTSP has multiple trucks to be managed at every iteration of a sequential constructive algorithm.

The authors make two main contributions towards establishing a DRL approach to min-makespan mTSP:

1- They propose a specialized graph neural network architecture which combines known ingredients in a way that is suitable to the structure of the mTSP;

2- They modify the RL training algorithm to take into account the intricate discrete structure of the makespan objective, which stabilizes the training process.

Experimentally, the proposed ScheduleNet method is trained on a single set of random instances with a small number of customers and trucks, then tested on similar and larger random instances, as well as some benchmark mTSP instances from the literature. Compared to some other learned and non-learned algorithms, ScheduleNet seems to be competitive.

Strengths:
1- Interesting engineering of the graph network model and of the RL training procedure to take into account mTSP and makespan structure;

2- Generalization from very tiny instances to much larger ones (though not too large in an absolute sense).

Weaknesses:
1- Experimental evaluation leaves many questions unanswered;

2- Motivation for tackling yet another variant of the TSP is not very strong, in that it is unclear that practitioners solving mTSP in practice would be interested in using the proposed method. No discussion of how the ideas presented here could extend to other variants of TSP or significantly improve performance on some class of instances that are of great interest to the community or an application domain.

3- Submission seems to have been rushed, with missing citations and weird captions in a couple places.

Recommendation: Overall, I have to recommend a rejection, but I do think that the authors are on a good path towards a paper if they strengthen the motivation and experiments. I don't know if that will be possible within the ICLR rebuttal.

Questions to the authors:

1- Table 1: how many instances are considered for each (N, m) pair here? Please provide standard deviations for the values provided here, without which it's hard to tell how stable the reported average makespan is.

2- Table 1: Why was CPLEX not run on the MTSP Uniform instances as was done in Table 2? This way you can compute the exact approximation ratio.

3- Table 1: Are the results reported for Hu et al. copied from that paper? If so, are you using the exact same set of graphs? If not, a direct comparison such as that claimed in Table 1 is not possible.

4- Table 2: The caption is hard to parse. The following does not make sense: "CPLEX results are reported as the average of the upper and lower bound." Instead, you should report the best solution found by CPLEX (i.e., the best upper bound at termination).

5- Table 2: Please define SOM, ACO and EA, and cite the respective papers as well as any additional implementation details.

6- OR-Tools: You should tune the parameters of OR-Tools (e.g., https://developers.google.com/optimization/routing/routing_options) on the same training set of instances that you train your model on. The tuning can be performed using some kind of grid search or more sophisticated tuning tools such as SMAC (https://github.com/automl/SMAC3).

7- Running time results: There is no mention whatsoever of the running time of ScheduleNet (in training, but more importantly at test time), and how it compares to OR-Tools and the other heuristics of Table 1-2.

8- ScheduleNet hyperparameters: You list the values but no mention of if/how they were tuned.

9- Relation to VRP: mTSP is a special case of VRP in which there are no worker (truck) capacities. Have you considered comparing ScheduleNet to VRP learning approaches from the literature, such as Nazari et al. (which you cite)?

Minor:
- "In this study, we formulate (MinMax mTSP as a Markov" --> remove "("
- Section 3.1, "Transition": I find this paragraph hard to parse.
- "is equals to the makespan" --> "is equal to the makespan"
- "since the following operations only for the given time step" --> "since the following operations only apply to the given time step"
- "instances whose number m of tasks and the number N of workers are sampled" --> shouldn't this be the opposite?
- "benchmark problems in mTSPLib (cite)," --> please add the appropriate citation: https://profs.info.uaic.ro/~mtsplib/
- Appendix: "and produces and produces “type-aware”"
- Appendix: "The computation steps of equation 11, 12,and ??"
- Appendix: "hyperparameters of SchduleNet" --> "hyperparameters of ScheduleNet"

---

> ### Author Response · Authors · 2020-11-24
> **Response to the reviewer 2**
>
> Thanks for the review. We revised the manuscript such that the questions are well answered.
>
> we have modified the introduction to further highlight the motivation and novelties of the current study. Please check the updated manuscript
>
> Answers for question 1: We repeat the experiments for each pair of $n$ and $m$ 500 times to secure the statistical supports on validating the performance of each method.  Additionally, we also report standard deviations for each pair of $N$ and $m$.
>
> Answers for question 2: In principle, solving the random mTSP instances with an exact method, such as formulating minmax mTSP as mixed-integer linear programming (MILP) and solve it with CPLEX is possible. However, we didn’t choose to solve the random instances since it usually takes a prohibitively large computation time, especially large instances $N\geq100$. Also, from the mTSP literature (not-necessarily deep + (RL) methods), we confirmed that the new algorithms are often compared to the existing heuristics.
>
> We choose to use OR-Tools for as the major target of comparisons since the OR-Tools often shows the reliable good outcomes for the Minmax mTSP problems where the optimal solution (or at least, the upper and lower bound are known cases).
>
> Answers for question 3: We delete the result of Hu et al. from the table. Instead, we additionally implement the heuristics and evaluate performances of the heuristics on the same mTSP instances to further investigate the properties of the proposed methods.  In summary, our method scores better-or-similar performance on various mTSP scenarios. And we confirmed that Schedulenet can win the OR-Tools on random mTSP dataset and in benchmark setup, Our method produces slightly large makespans but beats the meta-heuristic baselines which are optimizing the parameters for each instance.
>
> Answers for question 4: We replace the benchmark results of CPLEX with the so-far-known best upper bound.
>
> Answers for question 5: We update the manuscript to define the meta-heuristic algorithms, self-organization Map (SOM), ant-colony Optimization (ACO), evolutionary algorithm (EA), and add citations for the results.
>
> Answers Q6: We manually tuned the ‘search limits’ parameters of the OR-Tools described in the above link.
> Especially, we allows longer run times for solving large-sized maps to the OR-Tools. However, as we discussed in the manuscript, OR-Tools’ results consistently tend to be strong in smaller worker cases (m <=5) and weak in large worker cases (m=7).
>
> Answers Q7. Running time results. As per request, we included the computation time of ScheduleNet and comparison to OR-Tools in the Appendix. Empirical results show that ScheduleNet is faster, and its computation time is invariant to the topology (construction) of the underlying mTSP instance, and only depend on the problem size (m+N).
>
> Answers Q8: We deliberately choose the hyperparameters to become deep learning standards. e.g., $2^n$-numbered of neurons, ReLU activations, etc. Also, we set the hyperparameters for training and RL methods as similar to one of the stable-baseline PPO2, which is considered to be the standard PPO implementation now.  We included the hyperparameters and details of Training, Heuristics in the Appendix.
>
> Answers Q9. Relation to VRP... Have you considered comparing ScheduleNet to VRP learning approaches from the literature, such as Nazari et al. (which you cite)?
>
> Thank you for asking this question. We want to emphasize, that although mTSP is the special case of the CVRP (C = infinity), the problem formulations and approaches that were used by Nazari et. al. and Kool et. al. cannot be directly applied to solve minmax multi-worker (m>1) problems. Both Nazari et.al. and Kool et. al. mentioned that they are tackling CVRP problems with m=1, i.e. single-vehicle cases. Extending their works to multi-worker cases, was one of the main motivations of the ScheduleNet, and, indeed, we are planning to extend our approach to multi-vehicle capacitated VRP (mCVRP) and Flexible Job Shop Scheduling Problem (FJSSP).

---

> > ### Author Response · Authors · 2020-11-25
> > **Response to the reviewer 2 - Continued**
> >
> > 1- Experimental evaluation leaves many questions unanswered;
> > We agree that our first submission leaves questions a lot. From the revised manuscript, we did our best to answer the unanswered question. The major "answered" questions are as follows:
> >
> > 1. We clarify the set of instances where we are better than OR-Tools on the random uniform dataset. To summarize, ScheduleNet shows the cases winning the OR-Tools for every number of cities $N$ and the number of workers $m$  pairs. The winning ratio varies across the $N$ and $m$: ScheduleNet becomes stronger whenever the problems become larger.
> >
> > 2. We confirmed and provided the results about "scalability" in the computational time of the proposed method. In short, for large instances, our approach is faster than OR-Tools. This becomes fascinating since we also show that the proposed method most likely produce better solutions than OR-Tools.
> > We further analyze the effectiveness of the proposed return normalization scheme and RL training method. (1)
> >
> > 3. We show that the proposed return leads the RL agents to better performance than two dense rewards, which are practically considered to use training RL algorithms for solving scheduling problems. (2) Clipped REINFORCE, which can be seen as a simplified PPO, actually results in better outcomes than PPO. We hypothesize this phenomenon originates from the value prediction error of critics. We also provide the distribution of critic targets (makespan) with high volatility and even have multi-modalities, which make training critic challenging.
> >
> > Please refer to section 6 of the revised manuscript that thoroughly explains the "answers."
> >
> > 2- Motivation for tackling yet another variant of the TSP is not very strong, in that it is unclear that practitioners solving mTSP in practice would be interested in using the proposed method. No discussion of how the ideas presented here could extend to other variants of TSP or significantly improve performance on some class of instances that are of great interest to the community or an application domain.
> >
> > **Practical motivation**: Solving mTSP, especially when the number of cities $N$ and workers $m$, are practically demanding.  Let us think about the scenarios where a giant fully autonomous warehouse with the retrieving robots are in there. In such facilities, the number of items (naturally considered to be a city) and the number of retrieving robots (salesman) can be huge. In this practical setup, ScheduleNet can be better than the other baseline discussed in the manuscript.
> >
> > Also, this practical setup invokes a very different aspect of the mTSP problem. Unlike the problems we majorly discussed in the manuscript, the new tasks emerge during physically "solving" mTSP in the real world.
> >
> > This scenario is often referred to as "dynamic" scheduling problems. The proposed method (construction heuristics) is the only method that can solve such dynamic problems without resolving the whole new mTSP. The other methods, such as exact method or construction heuristics (e.g., OR-Tools and seq-to-seq RL methods), require solving the problem from scratch.
> >
> > **Extention to the general scheduling problems**: We plan to extend the focus of this research to more general scheduling problems, including:
> > 	1. the problems the AI community starts to deal with (e.g., Multi-robot reward collection (MRRC) and mCVRP).
> > 	2. the more practically demanding-yet-challenging problems such as Job-shop scheduling problems (JSSP); especially the flexible JSSP (FJSSP). (F)JSSP arises from various contexts. From daily-life usage: e.g., job scheduling in over the multiple CPU/GPU cores, to the high-tech industries such as semi-conduction manufacturing systems.
> >
> > We designed our problem the formulation and policy learning procedure being as much as independent from a specific type of scheduling problems such that, in future, the mentioned scheduling problems would be solved with the minimal change of method.
> >
> > 3- Submission seems to have been rushed, with missing citations and weird captions in a couple places.
> >
> > We apology for the first submission not tidy in terms of contents and grammar also. We wish that this revised manuscript can answer the questions made from the reviewing process.
> >
> > + We fixed the missing citations and the weird captions. Thank you for the comments.

---

### Official Review · AnonReviewer4 · 2020-10-29
**Insufficient performance and contribution**

**Rating:** 4
**Confidence:** 3

**Review:**

The authors propose an RL framework, called ScheduleNet trained by clipped REINFORCE, for minmax multiple traveling salesman problem (minimax mTSP), which uses a clipping idea to stabilize learning process as PPO does. The authors empirically show the feasibility of the proposed framework.

- Unfortunately, the proposed method has poorer performance than existing works, in particular, OR-Tool. This decreases the merit significantly.

- In addition, it is hard to find contribution from proposing new RL method since the stabilizing effect of the clipped REINFORCE is shown in only limited environment (only minimax mTSP).

- Table 2 is not completed.

- The nature of "minimax" mTSP should be more clearly represented and exploited. Currently, the proposed RL framework seems to work for other formation of mTSP, and also it seems not to exploit the nature of minimax.

- In order for showing novelty of the proposed method, it might be useful to devise and investigate actor-critic methods sharing the main idea. In the submission, only footnote 1 simply mentions the hardness of learning value function.

- The behavior of ScheduleNet need to be studied further, e.g., when your algorithm works well, and not.

---

> ### Author Response · Authors · 2020-11-24
> **Response to the reviewer 4**
>
> we have modified the introduction to further highlight the motivation and novelties of the current study. Please check the updated manuscript
>
> **[Unfortunately, the proposed method has poorer performance than existing works, in particular, OR-Tool. This decreases the merit significantly]**
>
> The proposed method cannot perform better than OR-tools on small to medium size instances. However, our approach shows a significant chance to win the OR-tools on the uniform random dataset of larger size, as provided in the updated manuscript. We kindly request to look at the revised manuscript.
>
> We would like to emphasize that the proposed approach tends to perform better for large-size problem instances with many vehicles and customers, as the search space becomes prohibitively larger for the meta-heuristic algorithms. We would like to highlight that the proposed method can find a reasonably good solution with less computational time than OR-tools due to the decentralized decision-making strategy. To prove this, we include the computation time curves (vs OR-tools) in the Appendix.
>
> **[In addition, it is hard to find contribution from proposing new RL method since the stabilizing effect of the clipped REINFORCE is shown in only limited environment (only minimax mTSP).]**
>
> Our main contribution is not on proposing new RL method but rather on proposing a decentralized sequential decision-making scheme to solve mTSP.  Specifically, we formulate mTSP as a semi-MDP and derive a decentralized decision making policy in a multi-agent reinforcement learning framework using only a sparse and delayed episodic reward signal. The major components of the proposed method and their importance are summarized as follows:
>
> -(Formulation)
> Decentralized cooperative decision making strategy: Decentralization of scheduling policy is essential to ensure the learned policy can be employed to schedule any size of mTSP problems in a scalable manner; decentralized policy map local observation of each idle salesman one of feasible individual action while joint policy map the global state to the joint scheduling actions.
>
> -(Forward propagation)
> State representation using type-award graph attention (TGA): the proposed method represents a state (partial solution to mTSP) as a set of graphs, each of which captures specific relationships among works, cities, and a depot. The proposed method then employs type-aware graph attention (TGA) to compute the node embeddings for all nodes (salesman and cities), which are used to assign idle salesman to an unvisited city sequentially.
>
> -(Learning)
> Training decentralized policy using a single delayed shared reward signal: Training decentralized cooperative strategy using a single sparse and delayed reward is extremely difficult in that we need to distribute credits of a single scalar reward (makespan) over the time and agents. To resolve this, we propose a stable RL training scheme which significantly stabilizes the training and improves the generalization performance.
> In short, the clipped Reinforce algorithm is designed specifically to train the ScheduleNet for solving mTSP.
>
> **[Table 2 is not completed.]**
> Both table 1 and table 2 are updated to compare the results thoroughly. We additionally provide heuristic baselines for further comparison. We kindly request the reviewers to check the revised manuscript.
>
> **[The nature of "minimax" mTSP should be more clearly represented and exploited. The proposed RL framework seems not to exploit the nature of minimax.]**
>
> The “max” component in the “minimax” problem is used just to compute the performance of a solution (the maximum traveling distance of all the vehicles).  Once this reward is computed as a reward signal, typical RL approach can be used to minimize this reward. In other words, there are no competitive relationships among multi-agents, the situation where a typical “minimax” game is seeking to address. In short, the “minimax” problem can be described as “minimizing the total completion time of all agents.”
>
> **[It might be useful to devise and investigate actor-critic methods sharing the main idea.]**
> Representing a function over combinatorial action space using deep NNs is a challenging task because the small change in the combinatorial input space can result in a significant change in the output. Also, the approximated function cannot be differentiated with the input, limiting the trained model to be used for decision making. Fitting the performance (makespan) that is severely nonlinear and (possibly) discontinuous using continuous function approximation will induce performance degradation.
>
> Due to these challenges, we do not fit the critic and only use the learned policy (actor). To empirically prove this argument, we conducted further extensive experiments showing that Clipped REINFORCE without Critic performs better than PPO  for both sparse and dense reward structures. We kindly request reviewers to look at the revised manuscript.

---

> > ### Author Response · Authors · 2020-11-24
> > **Response to the reviewer 4  - Continued**
> >
> > **[The behavior of ScheduleNet need to be studied further]**
> > We agree with that. We definitely add more detailed qualitative analysis in the revised manuscript.

---

### Official Review · AnonReviewer1 · 2020-11-01
**It proposes an RL algorithm to solve mTSP problem.**

**Rating:** 5
**Confidence:** 5

**Review:**

The mTSP with the goal of minimizing the longest route is considered, which minimizes the maximum route. This objective results in a balanced-length set of routes, which compared to the sum of routes obtains a more practical result. A graph representation based on the worker and assigned tasks is defined, then a type-aware graph attention (TGA) embedding procedure is proposed in which it obtains an embedding for node and edge representation. The state for each entity involves the 2D coordinates of the entity and the boolean indicator of idleness and assigned task of the worker. The action is worker-to-task assignment, and the reward is the makespan of finishing the problem, which is a sparse function. In type-aware graph attention (TGA) embedding, the embedding of node and edge are obtained and using the attention mechanism, an important weight for the embedding of each edge is obtained. The embedding functions are type-dependent which means the embedding considers the source node-type. The final message value for each node is obtained by multiplying the weight and edge embedding value of its neighbors. Then, an MLP is used to obtain a value for each source and possible target node, which takes the final value of the source, target, and edge between those as input. Then, the output of the MLP is used to get the final probability of choosing the next node.

Major comment:
The proposed algorithm looks interesting. Specifically, this problem can be modeled as multi-agent cooperative RL problem, and this paper suggests a model to handles it as a single RL problem. (For example, see paper [1] which suggests a multi-agent approach for a similar problem). However, the numerical results do not suggest a competitive algorithm yet and there are several baselines which obtain smaller objective in a smaller time, and yet it does not make sense to solve mTSP with a RL method. Note that this is not the case in VRP, since the current non-learning/non-commercial algorithms are not powerful in solving even medium-size problems with 50 nodes.

[1] Zhang, Ke, et al. "Multi-Vehicle Routing Problems with Soft Time Windows: A Multi-Agent Reinforcement Learning Approach." arXiv preprint arXiv:2002.05513 (2020).

+

minor comment:
The citation of mTSPLib is missing.

---

> ### Author Response · Authors · 2020-11-24
> **Paper updated with additional numerical results.**
>
> Thank you for reviewing and our paper. Please let us address your concerns.
>
> we have modified the introduction to further highlight the motivation and novelties of the current study. Please check the updated manuscript
>
> - ***Multi-agent cooperative RL problem ***\
> Indeed, our task can be viewed as a multi-agent cooperative RL problem. In this study, we propose a learning-based decentralized and sequential decision-making algorithm for solving Minmax mTSP problem in the MARL framework (centralized training and decentralized execution). The reason why we did not emphasize the MARL aspect is that when all agents share the same decision making policy through parameter sharing, it can be viewed as a single agent RL. However, each agent clearly makes independent action based on its own local observation in a decentralized manner.
>
> The trained policy, which is a construction heuristic, can be employed to solve mTSP instances with any number of salesman and cities. Learning a transferable mTSP solver in a construction heuristic framework is significantly challenging comparing to its single-agent variants (TSP and CVRP) because (1) we need to use the state representation that is flexible enough to represent any arbitrary number of salesman and cities (2) we need to introduce the coordination among multiple agents to complete the geographically distributed tasks as quickly as possible using a sequential and decentralized decision making strategy and (3) we need to learn such decentralized cooperative policy using only a delayed and sparse reward signal, makespan, that is revealed only at the end of the episode.
>
> To tackle such a challenging task, we formulate mTSP as a semi-MDP and derive a decentralized decision making policy in a multi-agent reinforcement learning framework using only a sparse and delayed episodic reward signal. The major components of the proposed method and their importance are summarized as follows:
>
> (Formulation) Decentralized cooperative decision-making strategy: Decentralization of scheduling policy is essential to ensure the learned policy can be employed to schedule any size of mTSP problems in a scalable manner; decentralized policy maps local observation of each idle salesman one of feasible individual action while joint policy maps the global state to the joint scheduling actions.
>
> (Forward propagation) State representation using type-award graph attention (TGA): the proposed method represents a state (partial solution to mTSP) as a set of graphs, each of which captures specific relationships among works, cities, and a depot. The proposed method then employs type-aware graph attention (TGA) to compute the node embeddings for all nodes (salesman and cities), which are used to assign idle salesman to an unvisited city sequentially.
>
> (Backward propagation) Training decentralized policy using a single delayed shared reward signal: Training decentralized cooperative strategy using a single sparse and delayed reward is extremely difficult in that we need to distribute credits of a single scalar reward (makespan) over the time and agents. To resolve this, we propose a stable MARL training scheme which significantly stabilizes the training and improves the generalization performance.
>
>
>
> -***Performance***\
> We updated Table 1 with additional "2phase" heuristics (K-Means Clustering + TSP Insertion Heuristics). Our results show our competitiveness in regards to the additional baselines. Also, we can observe that ScheduleNet performs comparatively or better than the OR-Tools on problems with a large number of workers.
>
>
>
> -***VRP***\
> Thank you for this suggestion.
>
> mTSP and VRP have different types of constraints. To solve mTSP, we need to consider the coordination among multiple agents who employing decentralized policies. To solve VRP, we need to consider a capacity constraint that requires more strategic routing behavior. Although RL-based approaches have been employed to solve VRP, there are few RL/MARL approaches to solve mTSP. We are planning to expand the mTSP problem to consider the vehicle capacity in the future. This type of problem can be perceived as a multi-agent vehicle routing problem with capacity constraints.
>
> In a similar direction, we are planning to expand our approach to solve the Flexible Job Shop Scheduling Problem (FJSSP) that also requires multi-robot coordination with joint constraints imposed on them.

---

### Decision · Program_Chairs · 2021-01-07
**Final Decision**

**Decision:**

Reject

**Comment:**

This paper proposes a deep reinforcement learning approach for solving minimax multiple TSP problem. Their main algorithmic contribution is to propose a specialized graph neural network to parameterize the policy and used a clipped idea to stabilize the training. Unfortunately, the reviewers remain to be unconvinced by the experiments after the rebuttal and the writing need to be significantly improved. Also, it would be worthwhile to study how the proposed method can generalize to other problems.